# Anti-Tuberculosis Potential of OJT008 against Active and Multi-Drug-Resistant Mycobacterium Tuberculosis: In Silico and In Vitro Inhibition of Methionine Aminopeptidase

**DOI:** 10.3390/ijms242417142

**Published:** 2023-12-05

**Authors:** Collins Onyenaka, Kehinde A. Idowu, Ngan P. Ha, Edward A. Graviss, Omonike A. Olaleye

**Affiliations:** 1Department of Pharmaceutical Sciences, College of Pharmacy and Health Sciences, Texas Southern University, Houston, TX 77004, USAkehinde.idowu@tsu.edu (K.A.I.); 2Center for Infectious Disease Research, Houston Methodist Research Institute, Houston, TX 77030, USA

**Keywords:** tuberculosis, *Mycobacterium tuberculosis*, methionine aminopeptidase (MetAP), N-terminal methionine excision (NME)

## Abstract

Despite the recent progress in the diagnosis of tuberculosis (TB), the chemotherapeutic management of TB continues to be challenging. *Mycobacterium tuberculosis* (*Mtb*), the etiological agent of TB, is classified as the 13th leading cause of death globally. In addition, 450,000 people were reported to develop multi-drug-resistant TB globally. The current project focuses on targeting methionine aminopeptidase (MetAP), an essential protein for the viability of *Mtb*. MetAP is a metalloprotease that catalyzes the excision of the N-terminal methionine (NME) during protein synthesis, allowing the enzyme to be an auspicious target for the development of novel therapeutic agents for the treatment of TB. *Mtb* possesses two MetAP1 isoforms, MtMetAP1a and MtMetAP1c, which are vital for *Mtb* viability and, hence, a promising chemotherapeutic target for *Mtb* therapy. In this study, we cloned and overexpressed recombinant MtMetAP1c. We investigated the in vitro inhibitory effect of the novel MetAP inhibitor, OJT008, on the cobalt ion- and nickel ion-activated MtMetAP1c, and the mechanism of action was elucidated through an in silico approach. The compound’s potency against replicating and multi-drug-resistant (MDR) *Mtb* strains was also investigated. The induction of the overexpressed recombinant MtMetAP1c was optimized at 8 h with a final concentration of 1 mM Isopropyl β-D-1-thiogalactopyranoside. The average yield from 1 L of *Escherichia coli* culture for MtMetAP1c was 4.65 mg. A preliminary MtMetAP1c metal dependency screen showed optimum activation with nickel and cobalt ions occurred at 100 µM. The half-maximal inhibitory concentration (IC_50_) values of OJT008 against MtMetAP1c activated with CoCl_2_ and NiCl_2_ were 11 µM and 40 µM, respectively. The in silico study showed OJT008 strongly binds to both metal-activated MtMetAP1c, as evidenced by strong molecular interactions and a higher binding score, thereby corroborating our result. This in silico study validated the pharmacophore’s metal specificity. The potency of OJT008 against both active and MDR *Mtb* was <0.063 µg/mL. Our study reports OJT008 as an inhibitor of MtMetAP1c, which is potent at low micromolar concentrations against both active susceptible and MDR *Mtb*. These results suggest OJT008 is a potential lead compound for the development of novel small molecules for the therapeutic management of TB.

## 1. Introduction

*Mycobacterium tuberculosis* (*Mtb*) is the causative organism of tuberculosis (TB) disease, and the tubercle bacilli was first discovered by Robert Koch (1843–1910) [1] and named *Mtb*, 1 year later [2]. According to the 2022 World Health Organization (WHO), TB is classified as the 13th leading cause of death globally. *Mtb* is one of the principal causes of a single infectious diseases, and second to Coronavirus disease 2019 (COVID-19) [3]. The WHO reported approximately 10 million people were infected with active tuberculosis in 2020 globally [3]. In 2022, 450,000 incidents of multi-drug-resistant/rifampicin-resistant TB (MDR/RR-TB) were recorded, of which, 78% were reported to be MDR-TB [4]. About 1.3 million deaths caused by TB were recorded in 2020, which is approximately twice the number of deaths caused by human immunodeficiency virus/acquired immunodeficiency syndrome (HIV/AIDS). The COVID-19 pandemic, in comparison to HIV/AIDS mortality, has critically impacted TB mortality in 2020 [3]. The eradication of TB disease with the current standard drug treatment has been challenging due to the rise in *Mtb* multi-drug-resistant strains [5,6].

After the United States Food and Drug Administration (FDA) approved rifampicin as a TB drug in 1971, there had been a lacuna in the discovery of new antimycobacterial. Recently, 43 antibiotics have been developed that were, in September 2020, in phase 1–3 of clinical trials and of which only 12 of the antibiotics targeted *Mtb* [4]. Therefore, the need to discover novel antimycobacterial agents to combat TB is critical due to the current drug-resistant TB cases and the longer duration of standard TB therapy regimen, 6–9 months, with its compliance challenges [7]. Hence, the discovery of new antimycobacterial with new mechanisms of action is pertinent to combat the case of antibiotic *Mtb* resistance.

To discover novel TB inhibitors and new chemotherapeutic targets, our projects has focused on methionine aminopeptidases (MetAP), a ubiquitous enzyme that has been validated as essential for the viability of *Mtb* [8]. This enzyme has been reported in other bacteria such as *Escherichia coli* (*E. coli*) [9] and higher vertebrates such as humans [10,11]. The essentiality of MetAP in all life forms has made it an auspicious target for discovering and developing new antibacterial agents, including drug-resistant TB [12]. MetAP is a dinuclear metalloprotease that functions in the excision of the N-terminal [11] methionine (NME) from nascent proteins and peptides emanating from ribosomes during translation [10]. N-formyl-Methionine is the amino acid residue that initiates the expression of polypeptides in bacteria, mitochondria, and chloroplast. It is usually the first amino acid to be removed from smaller N-terminal residues like valine, serine, threonine, cysteine, glycine, proline, or alanine [13,14]. NME is a co-translational regulation mechanism, universally essential in plants, bacteria, and vertebrates [11,15,16]. N-terminal methionine processing occurs in about 55–70% of nascent proteins [11]. NME is a crucial co-translational and post-translational pathway required for most proteins to undergo modifications such as subcellular localization, protein–protein interaction, stability, and protein degradation [11]. Studies have shown that the MetAP’s NME function is activated by divalent metal ions [10,17,18].

There are two types of MetAP, MetAP1 and MetAP2, and the difference between both classes lies in the presence of an inserted internal polypeptide in the catalytic domain sequence of MetAP2 [19]. MetAP1 has four subtypes, i.e., 1a, 1b, 1c, 1d, and 1n [20,21]. MetAP1b and MetAP1c are distinguished from MetAP1a by the presence of N-terminal extension in the subtypes. The N-terminal extension is vital for the activity of MtMetAP1c [22]. Elucidated X-ray structures of MetAP1c in both the apo and holo forms have revealed a conserved proline residue in the N-terminal extension. However, this conserved residue in MetAP1c is absent in MetAP1a, and the N-terminal extension of MetAP1c has 48% homology with the linker region of human MetAP1 [20]. The essential function of MetAP in universal cell growth was discovered through genetic studies [9,23], and the deletion of either MetAP1, or MetAP2, or both MetAP types produces a slow-growing yeast phenotype or death [20].

In addition, the critical physiological role of MetAP in bacteria has made it an important drug target for discovering novel antibacterial agents. *Mtb* differs from most prokaryotes in that *Mtb* possesses two MetAP1 subtypes: MtMetAP1a and MtMetAP1c. MtMetAP1a (from the mapA gene, Rv0734) and MtMetAP1c (from the mapB gene, Rv2861c) share 33% sequence identity [13] but differ by a 40 amino acid sequence present in the N-terminal of MtMetAP1c, which suggests its relevance in ribosomal interaction [20]. Recent studies have proven MtMetAP1c is also essential for *Mtb* viability [24,25,26]. Both MtMetAP1a and MtMetAP1c in their apoenzyme forms can be activated by divalent metal ions including Fe^2+^, Ni^2+^, Co^2+^, and Fe^2+^ [27,28], and it has been shown that Ni^2+^ and Fe^2+^ can increase the activity of the apoenzyme form [29,30].

Therefore, in this study, we cloned and overexpressed a recombinant MtMetAP1c, and the activating effect of cobalt and nickel ions on this enzyme was examined. Moreover, the effect of N-benzyl-5-chloro-N,6-dimethyl-2-pyridin-2-ylpyrimidin-4-amine (OJT008, CID: 2740173 (Figure 1)), a novel inhibitory compound, against the nickel- and cobalt-activated MtMetAP1c was investigated in an in vitro assay and further elucidated through a computation technique. Furthermore, the potency of OJT008 against replicating MDR-TB was investigated. Bioactivity data from our previous study demonstrated the potent antiproliferative effects of OJT008 in macrophages infected with *L. major* amastigotes and promastigotes at submicromolar concentrations, with no cytotoxicity against host cells [31]. The study further showed OJT008 significantly reduced the parasitic load with no evident toxicity in a preclinical in vivo model [31]. This potency of OJT008 against Leishmanial methionine aminopeptidase 1 prompted the selection of OJT008 in this study.

## 2. Results and Discussion

### 2.1. Cloning of MtMetAP1c and Gene Purification and Characterization of MtMetAP1c

The overexpression of MtMetAP1c was achieved by constructing a recombinant pET28a-MtMetAP1c using digestion enzymes (Figure 2 and Figure 3). The amplified gene produced a prominent band corresponding to the 858bp for MtMetAP1c, corresponding to the 1 kb DNA marker on an agarose gel (Figure 2). The recombinant gene was successfully used to transform *Escherichia coli* DH5 alpha-competent cells. A colony from the purified recombinant plasmid, 5:1 insert gene-to-vector molar ratio, was obtained with no mutations and successfully transformed into *E. coli* BL21 (DE3) competent cells (Figure 4A,B). The mini-pilot overexpression of transformed expression host cells showed an optimized expression at 8 h post-induction for cells transformed with pET28a-MtMetAP1c (Figure 5). The overexpressed protein was purified to near homogeneity with immobilized metal affinity chromatography using His Pur™ Ni-NTA Resin. The purified and N-terminally poly-His-tagged MtMetAP1c appeared as a band corresponding to the 32 kDa on Coomassie blue-stained SDS-polyacrylamide gel (Figure 6). The average yield of MtMetAP1c from 1 L of the BL21 (DE3) culture was 4.65 mg.

### 2.2. Determination of the Catalytic Effect of the Divalent Metal of Recombinant MtMetAP1c Activity

As shown in Figure 7, 10.48 µM is the concentration for the optimum activity of expressed MtMetAP1c. Studies have shown that purified metalloenzymes are activated by several divalent metals: Ni (II), Co (II), Fe (II), Zn (II), and Mn (II) [32,33]. Divalent metal ions directly participate in the removal of N-terminal methionine from nascent polypeptides by MetAP [34], and the activity of MetAP is inhibited by EDTA.

In this study, we screened for the activation of MtMetAP1c by the divalent metal ions, Co (II) and Ni (II). MtMetAP1c was discovered to be active in the presence of Ni^2+^ or Co^2+^ (Figure 8a,b). The study observed that both metals had their optimum catalytic effect at the same concentration of 100 µM and produced approximately the same activity of 258.8 µM mL^−1^ min^−1^ and 267.8 µM mL^−1^ min^−1^ for NiCl_2_ and CoCl_2_, respectively (Figure 8a,b). The efficiency of nickel ions in catalyzing MtMetAP1c is discovered to be insignificantly lower than that of the cobalt ions. Furthermore, this study showed that an increase in the concentrations of the two metals above 100 µM inhibits the activity of MtMetAP1c as evidenced by the decrease in the enzyme’s activity.

### 2.3. Determination of IC_50_ of OJT008 for MtMetAP1c Inhibition and Minimum Inhibitory Concentrations (MICs) of OJT008 in Replicating and Multi-Drug-Resistant Mtb

The inhibitory potential of OJT008 against NiCl_2_- and CoCl_2_ (100 µM)-activated MtMetAP1c was determined. The result showed OJT008 inhibited MtMetAP1c with IC_50_ values in the low micromolar range (Figure 9). The calculated IC_50_ values of OJT008 against CoCl_2_- and NiCl_2_ (100 µM)-activated MtMetAP1c are 11 µM and 40 µM, respectively (Table 1). OJT008 appears to be comparatively more potent against cobalt-activated MtMetAP1c compared to nickel-activated MtMetAP1c.

The MICs of OJT008 in replicating *Mtb* (CDC 1551 strain) and multi-drug-resistant *Mtb* (HN 3409 strain) were evaluated both for active and multi-drug-resistant *M. tuberculosis*. The MIC values of the compound against both *Mtb* phenotypes were less than 0.063 µg/mL (˂0.194 µM) and lower than the MIC values of 0.25 µg/mL and 5.00 µg/mL for isoniazid and kanamycin against CDC 1551 strain and HN 3409 strain, respectively. The MIC values of OJT008 are approximately more than 207-fold and more than 61-fold more potent than the in vitro inhibitory effect against 100 µM NiCl_2_- and CoCl_2_-activated MtMetAP1c, respectively (Table 2). However, the potency of the compound in *Mtb* (CDC 1551 strain) and multi-drug-resistant *Mtb* (HN 3409 strain) correlates with the inhibitory effect against MtMetAP1c in an in vitro assay as evidenced by a low IC_50_. The result suggests the compound to be more potent against the cellular *Mtb* in comparison to the in vitro MtMetAP1c activity. These values suggest OJT008 is a promising antimycobacterial agent.

### 2.4. Molecular Docking (In Silico) Analysis

The molecular docking analysis of OJT008 binding at the active site of MtMetAP1c with bound co-factors (cobalt and nickel) was performed to examine the OJT008 binding mode at the active site, and docking scores were estimated (Table 3). The docking score is a measure of the fitness of a ligand/drug into the binding pocket of a protein/enzyme, and the more negative the value, the better the fitness of a ligand/drug at the active site of the protein [35,36]. These scoring functions allowed the estimation and prediction of the binding pose and affinities of OJT008 at the binding pocket of the enzyme. The result showed an insignificant increase in the docking score of OJT008 with the enzyme in the presence of the nickel ion (−7.01 Kcal/mol) compared to the docking score in the presence of the cobalt ion (−6.52 Kcal/mol). These relatively high docking scores showed that OJT008 binds well with the enzyme in the presence of the two metal ions, and this further corroborated the inhibitory result observed in the in vitro study in the presence of the two ions.

The receptor–ligand plots of OJT008 were examined to elucidate the molecular interactions between OJT008 and the amino acid residues at the active sites and co-factors of the enzyme [37,38]. OJT008 was folded in the “pita-bread” shape, frequently seen in MetAP structures [10,27]. Figure 10 showed the 2-D visualization plot of the enzyme complexes. As shown in previous studies, many inhibitors of MtMetAP1c interacted with the divalent ions at the active sites of the enzyme [27,28,39], and in this study, OJT008 similarly interacted with both nickel and cobalt ions. The binding of OJT008 shows significant differences in its interactions with both nickel- and cobalt-bound structures, thereby suggesting different poses and a difference in docking scores. This might possibly explain the difference in IC_50_ in the ions-activated inhibitory assay (Table 1). The interactions of OJT008 with MtMetAP1c were compared to the interactions from an experimental study that evaluated the crystal structure of MtMetAP1c with an inhibitor,3-[(4-fluorobenzyl) sulfanyl]-4H-1,2,4-triazole (T03) [27]. Its results showed that OJT008 interacted not only with the co-factor (possibly chelating the ions) but with nearly similar amino acid residues (His100, Thr94, Cys105, His205, His202, Phe211, His114) as does the in vitro inhibitor, T03, as reported by Lu et al. [24], with similar types of interaction bonds such as hydrogen bond, p-sigma, p-cation, p-Sulfur, p-alkyl, p-p stacked interaction, donor–donor interactions, and Van der Waals (vdW) overlaps) observed. This finding further proves that OJT008 is a potent inhibitor of the MtMetAP1c enzyme.

## 3. Materials and Method

### 3.1. Materials

Middlebrook 7H9 broth, a *Mtb* culture medium, was purchased from Becton Dickinson. *Escherichia coli* BL21 DE3 expression host, pET28a vector, and *Escherichia coli* DH5 alpha competent cells were purchased from Invitrogen^®^ (Waltham, MA, USA). The restriction enzymes, BamhI and XhoI, were purchased from Invitrogen^®^, and the QIAGEN Plasmid Plus Midi Kit was purchased from Qiagen company (Hongkong, China). *Mtb* genomic DNA was purchased from ATCC^®^ (Manassas, VA, USA). HisPur™ Ni-NTA Resin and isopropyl beta-D-thiogalactopyranoside (IPTG) were purchased from Thermo Fisher Scientific^®^ (Waltham, MA, USA), and the OJT008 compound was purchased from Molport^®^ (Beacon, New York, NY, USA), and its dilutions were prepared in dimethyl sulfoxide (DMSO) purchased from Sigma Aldrich (St. Louis, MO, USA).

### 3.2. Method

#### Cloning of MtMetAP1c Gene and Expression of MtMetAP1c Protein

The protocol was adopted from the New England Biolab PCR Protocol for Taq DNA Polymerase with a standard Taq Buffer (M0273) (https://www.neb.com/protocols/0001/01/01/taq-dna-polymerase-with-standard-taq-buffer-m0273, accessed on 25 May 2022). The MtMetAP1c gene was amplified using the polymerase chain reaction (PCR) from the genomic DNA of *M. tuberculosis* H37Rv using the following forward and reverse primers, 5′-GCGGGATCCCCTAGTCGTACCGCGCTC-3′ and 5′-GCGTCGAGCTACAGACAGGTCAGGATC-3′, respectively. The PCR reaction was carried out in a 25 µL PCR mixture. BamhI and XhoI restriction enzymes were used to digest the DNA fragment, and the protein was cloned into the pET28a vector. The New England Biolabs (NEB) protocol was used for the double restriction digestion and ligation. The recombinant plasmid, pET28a-MtMetAP1c, was initially used to transform the non-expression host, *Escherichia coli* DH5 alpha competent cells, and the gene clone was extracted and purified from the non-expression host cell using a QIAGEN Plasmid Plus Midi Kit. The *E. coli* expression host, *Escherichia coli* BL21 DE3, was transformed with the clones of mutant-free recombinant plasmids for the overexpression of the MtMetAP1c protein. A single colony was grown overnight at 37 °C, 275 rpm, in 100 mL Listeria broth (LB) containing 30 µg/mL kanamycin. The following morning, the overnight culture was used to inoculate 2 L of LB containing 30 µg/mL kanamycin at 37 °C at the same speed of rotation until an OD_600_ of 0.6–0.8 was reached. The targeted protein was subsequently induced by the addition of 1 mM IPTG to a final concentration and further cultured at 16 °C for 8 h at 280 rpm before harvesting the cells. The 8 h post-induction time point was selected for the overexpression based on preliminary mini-pilot overexpression data using 5 mL of LB. The post-induction cell collection was taken at three different periods: 2, 4, and 8 h.

The optimum expression of cells was qualitatively determined using the sodium dodecyl sulphate–polyacrylamide gel electrophoresis (SDS-PAGE). The harvested cells were suspended in 1X PBS and sonicated in a lysis buffer (0.4% Triton X-100, 1xPBS, 10% glycerol, EDTA-free protease inhibitor tablets, and 20 mM imidazole). The cell lysate was centrifuged at 5000× *g* for 1 h, and the clarified protein was retrieved for the purification process.

The clarified protein obtained was incubated for 1 h with a pre-equilibrated His Pur™ Ni-NTA Resin, and then, the beads were washed for the first and second time with wash buffer (50 mM HEPES (pH 8.0), 100 mM NaCl, 10% glycerol, and 20 mM imidazole), and then, finally washed the third time with wash buffer (50 mM HEPES (pH 8.0), 100 mM NaCl, 10% glycerol, and 40 mM imidazole) and eluted with elution buffer containing 75 mM imidazole. The efficiency of the expression and purification protocol of the protein was determined by running SDS-PAGE on the cell lysate and the supernatant. The average yield, quantified with the Thermo Scientific NanoDrop UV-Vis spectrophotometer, was 4.65 mg of mtMetAP1c per liter of *E. coli* culture.

### 3.3. Determination of the Catalytic Effect of the Divalent Metal of Recombinant MtMetAP1c Activity

Initially, the activity of eluted MtMetAP1c was determined. An enzymatic assay screen to determine the NME ability of the cobalt ion-activated MtMetAP1c and the concentration of the protease required for optimum activity was performed. The enzymatic assay was a slight modification of the method adopted by Zhou et al. and Olaleye et al. [40,41], using a monopeptide substrate and the assay’s incubation time was 1 h. The reaction was performed at 37 °C in a 96-well plate, and the activity was monitored on a spectrophotometer at 405 nm. Each well had a total of 100 µL reaction containing 50 mM HEPES buffer (pH 8), 100 mM NaCl, 15 µM CoCl_2_, 100 µg/mL BSA, 1 mM chromogenic substrate (L-Methionine para nitroanilide), with a concentration range of 4.19, 8.38, 10.48, 12.58, 16.77, and 20.96 µM of MtMetAP1c. A unit of enzyme activity was defined as 1 µM of the substrate product, nitroanilide per minute under the stipulated experimental condition. The absorbance was measured with the spectrophotometer at 405 nm after 1 h of incubation. After validating that the expressed MtMetAp1c was active, the enzyme-activating effect of the two studied MtMetAp1c cofactors, cobalt (II) and nickel (II) ions, was accessed, adopting the same colorimetric assay. The following concentration range of CoCl_2_ and NiCl_2_ concentration were utilized for the enzymatic assay: 0.1, 1, 10, 100, 1000 µM, and 10 mM. The specific divalent metal concentrations that catalyzed the highest activity of MtMetAP1c were selected for the subsequent in vitro compound inhibitory effect screens.

### 3.4. Determination of IC_50_ of OJT008 for MtMetAP1c Inhibition

The half-maximal inhibitory concentration (IC_50_) of OJT008 for MtMetAP1c was evaluated. Based on the optimum activity obtained from the enzyme using nickel chloride and cobalt (II) chloride, MtMetAP1c was activated with 100 µM CoCl_2_ and 100 µM NiCl_2_. A 50 mM stock of the test compound in DMSO was prepared and serially diluted in DMSO to obtain a final concentration of 500 µM to 1 µM in each assay well of a 96-well plate. The reaction was carried out in triplicate. The total volume of assay per well was 100 µL containing a final concentration of 100 mM NaCl, 50 mM HEPES buffer (pH 8), 100 µg/mL BSA, 15 µM CoCl_2_, and 1 mM substrate. The compound was first preincubated with 10.48 µM MtMetAP1c for 20 min at 37 °C, and then subsequently incubated with the substrate for another 1 h at the same temperature, and then monitored at a wavelength of 405 nm on a spectrophotometer. Drug-free and DMSO media were used as controls. The background hydrolysis was corrected, and the data were analyzed with four-parameter (variable slope) equations using GraphPad Prism (Boston, MS, USA) version 6.0.

### 3.5. Determination of Minimum Inhibitory Concentration of OJT008 in Replicating Mtb

The screening pf the potency of OJT008 against active *Mtb* CDC 1551 strain was performed at a starting concentration range of 5 mg to 0.0125 mg/mL. The compound was serially diluted in DMSO and added to 0.2% glycerol, 0.05% Tween-80, and 10% albumin/dextrose complex (ADC) to obtain a final concentration of 50 to 0.125 µg/mL. The culture of *Mtb* CDC 1551 was grown to an OD of 1.0 and diluted to 1/100. A 0.1 mL volume of the bacterial culture was added to each drug dilution to obtain a total volume of 5 mL. The controls used for the screening of OJT008 against the replicating *Mtb* type included DMSO for a negative control, isoniazid for a positive control, and a drug-free medium as a blank. The *Mtb* CDC 1551 culture was observed for 14 days.

### 3.6. Determination of Minimum Inhibitory Concentration of OJT008 in Multi-Drug-Resistant Mtb

The screening of the potency of OJT008 against active *Mtb* HN 3409 strain was performed at a starting concentration range of 5 mg to 0.0125 mg/mL. The compound was serially diluted in DMSO and added to 2% glycerol, 0.05% Tween-80, and 10% albumin/dextrose complex (ADC) to obtain a final concentration of 50 to 0.125 µg/mL. The culture of *Mtb* HN 3409 was grown to an optical density (OD) of 1.0 and diluted to 1/100. A 0.1 mL volume of the bacterial culture was added to each drug dilution to obtain a total volume of 5 mL. The controls used for the screening of OJT008 against the multi-drug-resistant *Mtb* type included DMSO as a negative control, 5–0.25 mg/mL kanamycin as a positive control, and a drug-free medium as a blank. The *Mtb* HN 3409 culture was monitored for 3–4 weeks.

### 3.7. Molecular Docking

#### MtMetAP1c Acquisition and Preparation

X-ray crystal structures of nickel-ion- and cobalt-ion-bound MtMetAP1c (PDB code: 3IU8 and PDB code: 1YJ3) were obtained from the RSCB Protein Data Bank [20,27]. The structures were then prepared on the UCSF Chimera version 1.14 (San Francisco, CA, USA) software package [42]; the proteins were prepared by removing water molecules and performing nonstandard naming and protein residue connectivity as described by Uhomoibi et al. [43]. The 3-D structure of OJT008 was prepared on the Avogadro (Pittsburgh, PA, USA) version 1.97.0 package [44]. AMBER forcefield 14SB inbuilt on Chimera 1.14 (San Francisco, CA, USA) software was utilized for ligand minimization.

Molecular docking analysis was conducted as described by Idowu et al. [45]. Autodock vina available on Chimera version 1.14 (San Francisco, CA, USA) was used for molecular docking analysis [46], with default docking parameters. Before docking, Gasteiger charges were added to the molecules, and the non-polar hydrogen atoms were merged with carbon atoms. The molecules were then docked into the proteins’ binding pocket, by defining the grid box with a spacing of 1 Å for each molecule and (18 × 31 × 21) and (21 × 27 × 17) in size, pointing in x, y, and z directions, respectively. The protonation state of the target was performed before docking calculations. Exhaustiveness number eight was used. The molecular docking scores of OJT008 for both nickel and cobalt ions were estimated.

## 4. Conclusions

This study shows a promising potency of OJT008 against the MDR-TB, thereby strategically positioning OJT008 as a promising antimycobacterial pharmacophore against TB cases resistant to first-line TB treatment. The significance of our study is the potential characterization of a novel chemotherapeutic agent against active and MDR *Mtb*, thereby overcoming the challenges of the current TB drug regimen. In addition, the compound’s utilization of MtMetAP1c as a potential chemotherapeutic target provides the benefits of circumventing the global challenges posed by multi-drug-resistant (MDR) and replicating TB. Therefore, OJT008 can serve as a propitious lead in the discovery and development of novel antimycobacterial agents. This compound could be promising in the treatment of TB and could serve as a lead compound in the development of inhibitors of MtMetAP1c upon additional in vivo and clinical evaluations.

## Figures and Tables

**Figure 1 ijms-24-17142-f001:**
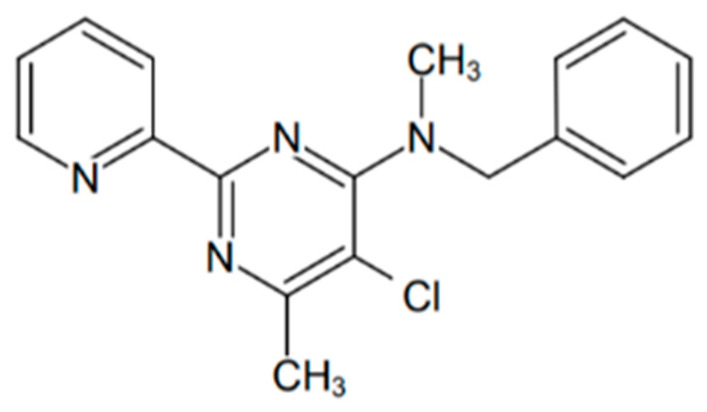
2-D structure of 0JT008.

**Figure 2 ijms-24-17142-f002:**
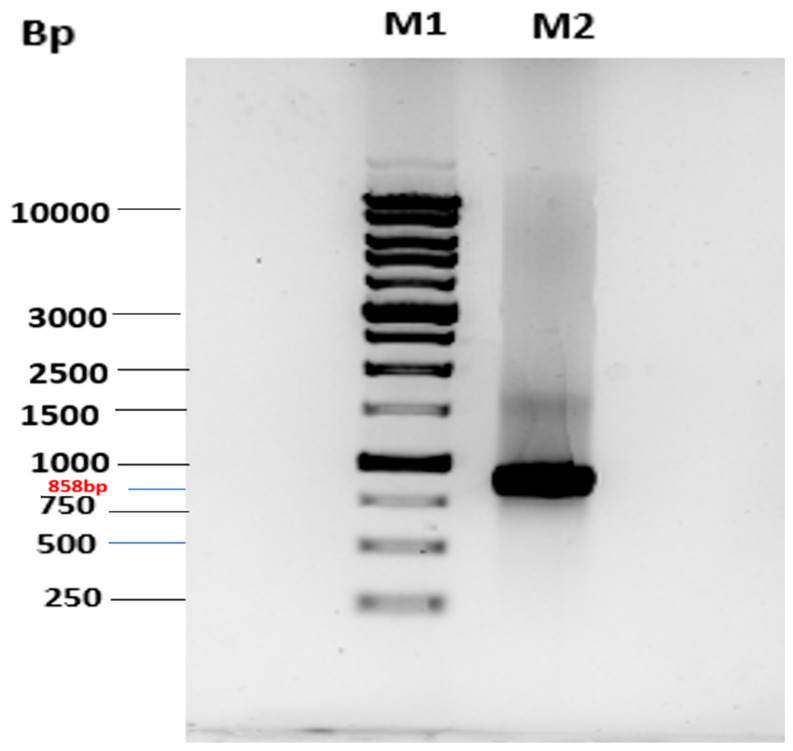
Agarose gel scan showing the PCR product. Lane 1 (M1): 5 µL of 1 Kb DNA ladder. Lane 2 (M2): amplified MtMetAP1c gene.

**Figure 3 ijms-24-17142-f003:**
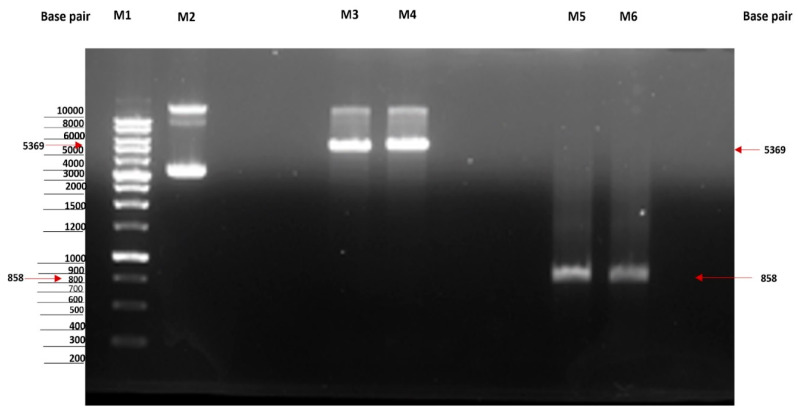
Agarose gel scan showing control and digested genes. Lane 1 (M1): 5 µL of 1 Kb DNA ladder. Lane 2 (M2): undigested vector pET28a (0.5 µg). Lanes 3 and 4 (M3 and M4): digested vector (0.5 µg each). Lanes 5 and 6 (M5 and M6): digested amplified target gene (0.5 µg each).

**Figure 4 ijms-24-17142-f004:**
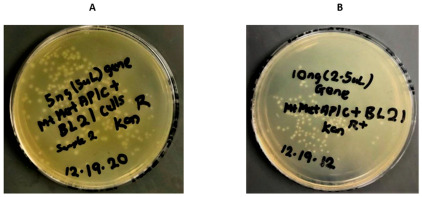
Transformed *Escherichia coli* BL21 (DE3) competent cells with 5 ng and 10 ng 5:1 molar ratio of the recombinant plasmid. (**A**) Competent cells transformed with 5 ng recombinant plasmid. (**B**) Competent cells transformed with 10 ng recombinant plasmid.

**Figure 5 ijms-24-17142-f005:**
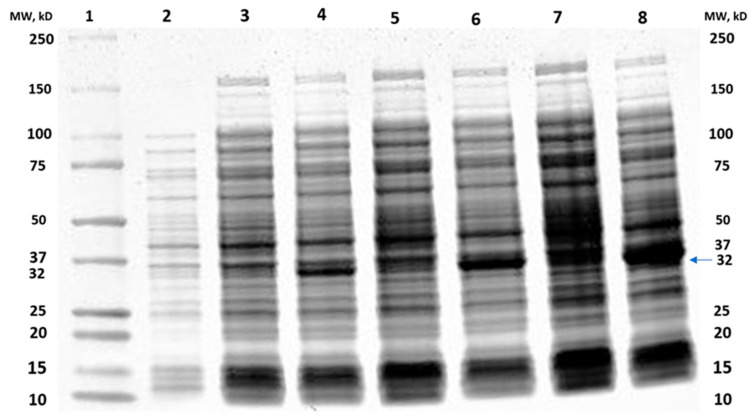
Mini induction pilot of recombinant MtMetAP1c from M. tuberculosis at different time intervals. **Lane 1,** molecular weight marker; **Lane 2**, uninduced whole cell lysate; **Lane 3**, uninduced cell lysate at 2 h; **Lane 4,** induced cells at 2 h; **Lane 5,** uninduced cells at 4 h; **Lane 6,** induced cells at 4 h; **Lane 7,** uninduced cells at 8 h; and **Lane 8,** induced cells at 8 h.

**Figure 6 ijms-24-17142-f006:**
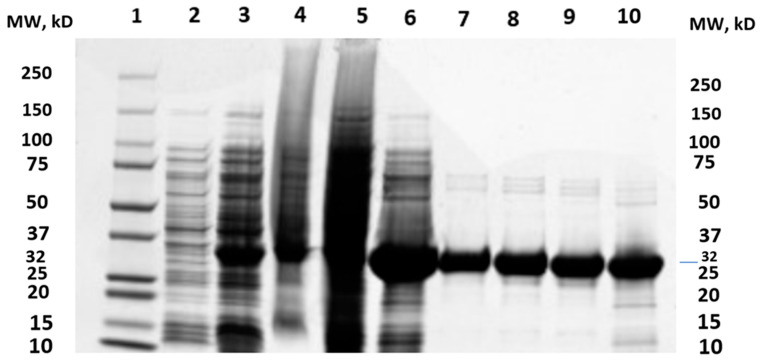
SDS–PAGE analysis of purified His-tagged MtMetAP1c protein using Ni–NTA-affinity chromatography. **Lane 1**, molecular weight marker; **Lane 2**, uninduced whole cell lysate; **Lane 3**, induced cell lysate taken after 8 h; **Lane 4**, pellet after sonication; **Lane 5**, supernatant containing clarified protein after sonication; **Lane 6**, beads after final binding; **Lane 7**, beads after final elution; **Lane 8**, elution 1 (using 75 mM imidazole); **Lane 9**, elution 2 (using 75 mM imidazole); and **Lane 10**, elution 3 (using 75 mM imidazole).

**Figure 7 ijms-24-17142-f007:**
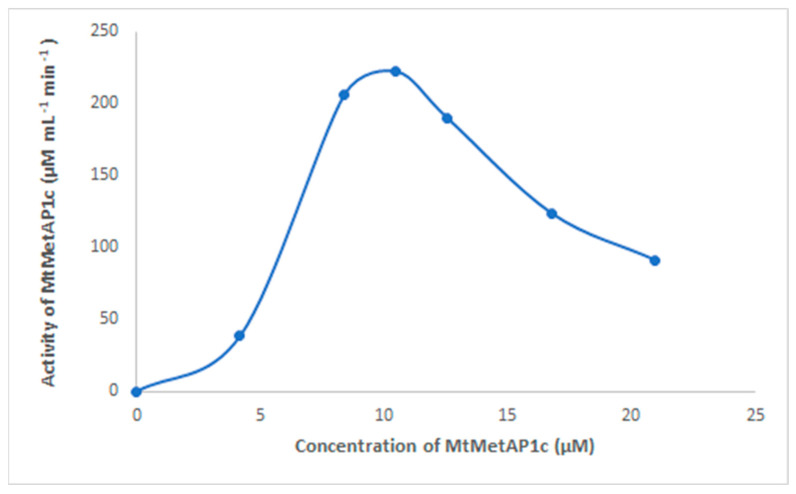
MtMetAP1c activity calibration curve, and the concentration (µM) of MtMetAP1c is the *x*-axis.

**Figure 8 ijms-24-17142-f008:**
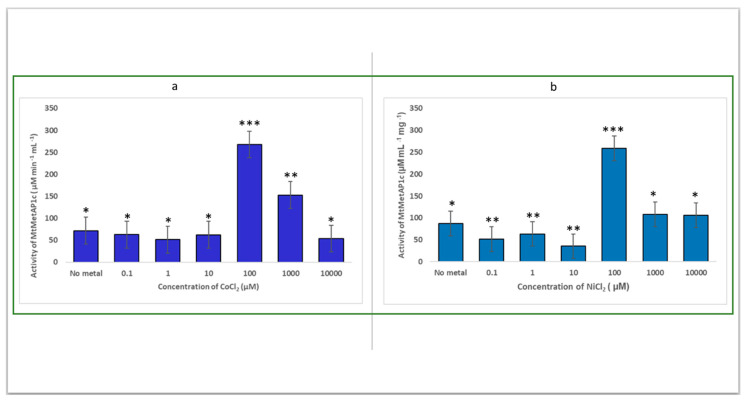
Activity of MtMetAP1c in the presence of different concentrations of (**a**) cobalt (II) chloride and (**b**) nickel (II) chloride. * means significant and insignificant difference (* *p* < 0.05, ** *p* < 0.01, *** *p* < 0.001).

**Figure 9 ijms-24-17142-f009:**
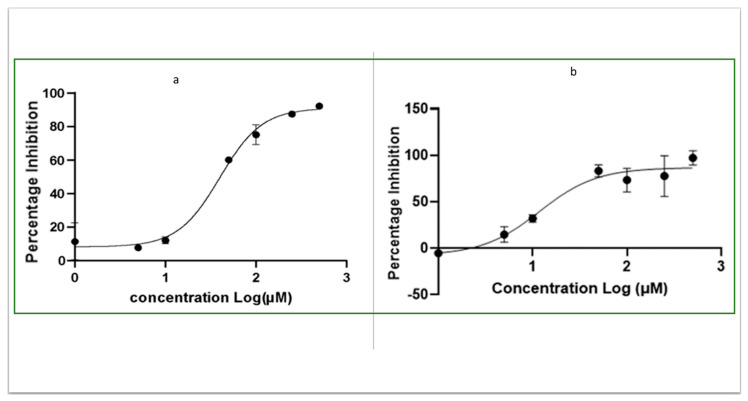
Effect of OJT008 against the activity of divalent metal-activated MtMetAP1c: (**a**) activity of 100 µM cobalt (II) chloride-activated MtMetAP1c and (**b**) activity of 100 µM nickel (II) chloride-activated MtMetAP1c.

**Figure 10 ijms-24-17142-f010:**
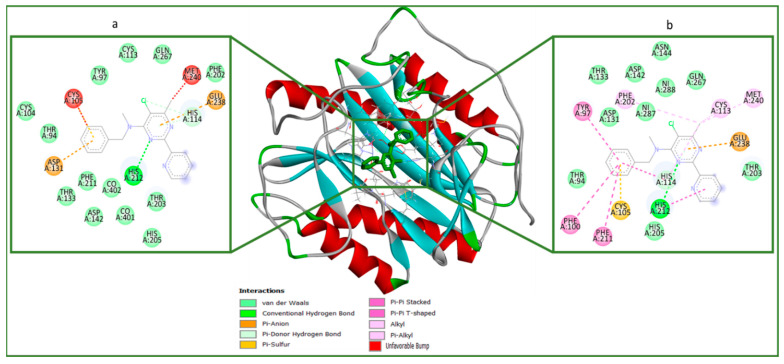
Representation of protein (MtMetAP1c)–OJT008 interactions plots with different bound divalent metals: (**a**) CoCl_2_ and (**b**) NiCl_2_ bound to MtMetAP1c.

**Table 1 ijms-24-17142-t001:** IC_50_ of OJT008 against metal-activated MtMetAP1c.

Inhibitor	IC_50_ (µM)	
	MtMetAP1c + 100 µM CoCl_2_	MtMetAP1c + 100 µM NiCl_2_
0JT008	11.81 ± 1.30	40.12 ± 1.12

**Table 2 ijms-24-17142-t002:** MICs of OJT008 in replicating and multi-drug-resistant *Mtb*.

Inhibitor	MIC (µg/mL)	
	Replicating *Mtb* (CDC 1551)	Multi-Drug-Resistant *Mtb* (HN 3409)
Isoniazid	0.25	ND
Kanamycin	ND	5.00
0JT008	<0.063	<0.063

**Table 3 ijms-24-17142-t003:** Molecular docking score of OJT008 towards the active site of metal-bound MtMetAP1c enzyme.

	Co-Enzyme	
	MtMetAP1c + CoCl_2_	MtMetAP1c + Ni Cl_2_
Docking Scores (Kcal/mol)	−6.52	−7.01

## Data Availability

All data are presented in the manuscript.

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
