# Peer review of "Anti-Tuberculosis Potential of OJT008 against Active and Multi-Drug-Resistant Mycobacterium Tuberculosis: In Silico and In Vitro Inhibition of Methionine Aminopeptidase"

_ijms, 2023, doi:10.3390/ijms242417142_

Round 1
Reviewer 1 Report
Comments and Suggestions for Authors
Dear authors, your work shows interesting results in the search for new antibiotics against M. tuberculosis. I send my respectful comments on your work.
Indicate the purity of OJT008.
Provide chemical data on OJT008, its IUPAC name, CAS, etc. bioactivity background and the rationale for selecting it.
In the materials section there are two subtitles indicating "Molecular Docking".
OJT001 was not studied, but is listed in materials.
Indicate the concentration of DMSO used in the experiments.
Figure 8 is wrongly cited in the text, it is figure 9.
It is not clear what is the relationship that the authors establish between the 100 uM concentration of the cations that were determined in vitro with the enzyme activity in the cellular context, i.e. are these concentrations achievable inside the bacteria?
Clarify the text of line 316-319.
Table 2 should be placed after the presentation of the docking results.
Since the authors select Vina as the program for the study of possible ligand interactions with the active site, it is essential to demonstrate that the program succeeds at least in reproducing the crystallographic pose of the co-crystallized ligand.
Explain in detail how the protonation state of the protein was calculated, which is critical especially considering the presence of histidine in the active site.
The authors should explain very clearly how the targets were prepared and how each of the cations was included in the complexes.
The authors indicate that the ligands have "relatively high docking scores" however, these are not exactly striking values on the autodock Vina scale.
The 2D plots of the complexes in the presence of Ni and Co show significant differences in the interactions, suggesting different poses, a result that should be expanded and explained by the authors. In addition, a 3D plot of each complex could be useful.
The authors should discuss the role of cations for ligand-target interactions, since as can be inferred from the 2D diagrams, they are apparently of marginal relevance. For this reason, redocking of the co-crystallized ligand is a necessary control to check that Vina succeeds in reproducing this type of interactions.
Comments on the Quality of English Language
Minor corrections are required.
Author Response
The authors appreciate the constructive contributions of the referee. Kindly find the details of the authors response attached.
Thanks.

Reviewer 2 Report
Comments and Suggestions for Authors
In this manuscript entitled “ Anti-tuberculosis Potential of OJT008 against Active and Multi-drug resistant Mycobacterium tuberculosis: In silico and In vitro inhibition of Methionine Aminopeptidase”, Onyenaka C. et al. reported a protocol to overexpressed MtMetAP1c, an essential metalloprotease in Mycobacterium tuberculosis (M.tb), and determined its in vitro activity in presence of different ions and the inhibitors OJT008. The authors also tested the susceptibility of OJT008 against replicating M.tb strains. From my understanding, this manuscript is a follow-up study of previous manuscripts published by the authors. However, I would advise the authors to revise the manuscript to include or detail the rationale of the manuscript and its primary aim. Also, I have some significant comments according to the results obtained from the manuscript, as listed below.
Major comments:
1-lines 64-67: The authors reported that MetAP is essential in M.tb and ubiquitous in several other pathogens, plants and humans. In my opinion, the statement that MetAp is an auspicious target does not favour the development of new inhibitors targeting MetAP, as potential side effects can rise due to the presence of the antimicrobial target in the host. I am puzzled by this statement and would believe that studies had been done to show that novel inhibitors like OJT008 have specificity for the mycobacterial MetAP1c. The authors must report or perform any cytotoxicity assays that were performed to show the selectivity of their inhibitor for bacteria and not humans.
2- Figure 8: the authors reported that the catalytic activity of the purified MetAP1c was increased in the presence of Cobalt and Nickel. However, the activity was not reported with other ions with a control. From this Figure, it is impossible to determine if the catalytic activity increases compared to other ions like Fe2+. Also, as reported in the introduction, this enzyme can use several ions. Is there one ion that is most likely used in mycobacteria to catalyze the activity of this enzyme? Will it be more relevant for in vitro testing to use such ions instead of others (like Ni and Co) that are only found in small amounts inside the bacteria?
3- Table 1: The authors reported the MIC of OJT008 in one replicating strain (CDC1551) and one multi-drug resistant isolate (HN3409). I would be cautious in writing that this compound works in replicating and MDR-TB isolate. One isolate is not representative. Also, the author didn’t report the MIC of rifampicin and isoniazid for the clinical isolate. So, confirming that the isolate is indeed and MDR-TB strain is impossible. Do the authors have the genotypic or phenotypic data to support that this isolate is MDR? If not, the authors must test the MIC and show that CDC1551 is susceptible to RIF and INH.
Minor comments:
1- Lines 55-56: can the authors revise the citation used for this sentence? I don’t think the WHO TB report from 2020 will report the number of drugs in development as of September 2020.
2- Line 100: Fe2+ is indicated twice in the sentence.
3- Line 108-112: No figure legend for the structure is presented.
4- Line 199: is there a reason why the MIC was determined in media containing 2% glycerol? This is relatively high, considering that the standard 7H9 complete media usually contains 0.2% glycerol.
5- Figure 4: I am unsure what the authors want to show on these Agar plates. Also, I would suggest taking the picture without the writing obstructing the colonies.
6- Figure 5: Did the authors load the same amount of proteins for each lane? How were the samples normalized? The total amount of proteins seems to increase with the induction time.
7- Figure 7: The calibration curve was performed in the presence of which ion? Should there be a calibration curve for each ion tested?
8- Line 301: should the authors refer to Figure 9 instead of Figure 8.
9- Table 2: Did the authors calculate the docking score in the presence of other ions? The authors should calculate according to the natural ions this enzyme can bind for its catalytic activity as a control.
Comments on the Quality of English Language
I would suggest the authors do spellcheck and minor revisions throughout the manuscript.
Author Response

(The authors gave the same response as above.)

Round 2
Reviewer 1 Report
Comments and Suggestions for Authors
The concerns and recommendations have been taken into account by the authors.
Author Response
The authors appreciate the constructive contributions of the referee.
Reviewer 2 Report
Comments and Suggestions for Authors
I would like to thanks the authors for answering my comments. However, depite their answer some of the issues were not directly address or modify in the manuscript. These issues were raised as it is not clearely indicated in the manuscript. Please revised in the manuscript my comments about Figure 5 and Figure 7.
One of my main concerns still remained even after revision by the authors. The authors claimed that OJT008 is active against multi-drug resistant Mycobacterium tuberculosis (as mentionned in the title of this manuscript). Despite the fact that the authors mentionned that they have phenotypic and genotypic evidence to support this claim for strain HN 3408 (MDR-TB isolate), the phenotypic results were not added to the masnucript. By definition (WHO), MDR-TB is defined by resistance to rifampicine and isoniazid. In Table 2, we can see that the MIC of isoniazid is ND (which I hypothetize that it means that it was "not determined" as this abbreviation was not defined). and the MIC to rifampicin is not reported. I will highly recommend that the authors modify their statement as it is contradictory to their comments.
Author Response
Referee’s comment
Figure 5: Did the authors load the same amount of proteins for each lane? How were the samples normalized? The total amount of proteins seems to increase with the induction time.
Authors’ Response:
An equal amount of protein was loaded into each well, 33.4 µg (10.48 µM) MtMetAP1c.
The total protein samples were normalized using the Bio-Rad Image Lab™ Software 2.0.
Yes, the total amount of proteins increased with an increase in the induction time.
Referee’s comment
Figure 7: The calibration curve was performed in the presence of which ion? Should there be a calibration curve for each ion tested?
Authors’ Response:
The calibration curve was performed in the presence of only Cobalt ions.
Referee’s comment
One of my main concerns still remained even after revision by the authors. The authors claimed that OJT008 is active against multi-drug resistant Mycobacterium tuberculosis (as mentioned in the title of this manuscript). Despite the fact that the authors mentioned that they have phenotypic and genotypic evidence to support this claim for strain HN 3408 (MDR-TB isolate), the phenotypic results were not added to the manuscript. By definition (WHO), MDR-TB is defined by resistance to rifampicine and isoniazid. In Table 2, we can see that the MIC of isoniazid is ND (which I hypothetize that it means that it was "not determined" as this abbreviation was not defined), and the MIC to rifampicin is not reported. I will highly recommend that the authors modify their statement as it is contradictory to their comments.
Authors’ Response:
The authors appreciate the contributions of the referee. Antimicrobial Susceptibility Tesing (AST) against the HN3408 Mycobacterium tuberculosis (MTB) strain has been done previously. Antimicrobial Susceptibility Tesing (AST) for the 4 primary first line were performed per protocol, using BACTEC 460 TB system. The first line drugs were provided in a drug kit (Becton Dickinson Laboratory, Cockeysville, Md.) with the following concentrations used: rifampicin (RIF) 2.0 μg /mL, isoniazid-low (INHL) 0.1 μg/mL, isoniazid-high (INHH) 0.4 μg / mL, pyrazinamide (PZA) 100 μg/mL and ethambutol 2.5 μg/mL. Briefly, drug susceptibility testing was done using standard procedure whereby 0.1 mL of the appropriate drug solution was injected into labeled 12 B vials which resulted in the desired concentration of a drug in the medium.
Susceptibility was reported as susceptible (S) or resistant (R).
In the case of HN3408, the phenotype was reported as:
rifampicin (RIF) - R
isoniazid-low (INHL) - R
isoniazid-high (INHH) – R
pyrazinamide (PZA) – S
ethambutol (ETH) - S